

# Internal and marginal fit of digitally fabricated all-ceramic crowns with auxiliary retentive features on short clinical abutments: a micro-CT study

Saeed M. Alqahtani[1,*], Saurabh Chaturvedi[1,2,3,*], Mohamed Khaled Addas[1], Nasser M. Alqahtani[1], Mohammad A. Zarbah[1], Mohammed Hussain Dafer Al Wadei[4], Feras Ali Alsaeed[1], Yasir Saad AlJaadan[1], Ali Abdullah Ali Alqahtani[1], Mohammed Abdullah Al Mansooi[5] and Mudita Chaturvedi[2]

[1] Department of Prosthetic Dentistry, College of Dentistry, King Khalid University, Abha, Aseer, Saudi Arabia
[2] Department of Dental Research Cell, Dr. D. Y. Patil Dental College and Hospital, Pune, Maharashtra, India
[3] Department of Prosthetic Dentistry, SPDC, DMIHER (DU), Wardha, Maharashtra, India
[4] Department of Restorative Dental Science, College of Dentistry, King Khalid University, Abha, Aseer, Saudi Arabia
[5] Department of Prosthodontic, Hamad Dental Hospital, Ministry of Health, Doha, Qatar
[*] These authors contributed equally to this work.

Corresponding author
Saurabh Chaturvedi,
survedi@kku.edu.sa

## ABSTRACT

**Background**. Short clinical crowns/abutments (SCC) pose a challenge in achieving adequate retention. Auxiliary retentive features (ARF), such as grooves, are commonly employed to enhance retention. The marginal gap (MG) and internal fit (IF) of restorations are critical factors influencing clinical success. This study aimed to evaluate the MG, IF, and cement volume of digitally fabricated (computer-aided design (CAD) and computer-aided manufacturing (CAM)) all-ceramic crowns with and without grooves on SCC using micro-computed tomography (micro-CT).

**Methods**. A mandibular second molar typodont tooth was prepared to simulate SCC. Four groups of crowns ($n = 30$ per group) were fabricated: Group 1: CAD/CAM zirconia without grooves, Group 2: CAD/CAM zirconia with grooves, Group 3: CAD/CAM lithium disilicate with grooves, Group 4: Conventional lithium disilicate with grooves. Crowns were cemented using resin cement. MG, IF, and cement volume were evaluated using micro-CT. Gap measurements were taken in two planes across seven zones (Z1 to Z7). Statistical analysis was performed using one-way ANOVA and *post hoc* Tukey tests.

**Results**. Significant differences in MG and IF were observed among the groups ($p < 0.001$). Group 1 exhibited the lowest MG and IF, followed by Group 3, Group 2, and Group 4. Group 4 showed the highest average marginal discrepancy (AMD) and average wall discrepancy (AWD) ((AMD: Z1 = 197.36 ± 10.56 μm; Z7 = 226.5 ± 8.24 μm), (AWD: Z2 = 150.05 ± 10.89 μm; Z6 = 169.38 ± 10.57 μm)), followed by Group 2 > Group 3 > Group 1. The greatest discrepancy at the cuspal area was observed in Group 2, followed by Group 3 > Group 1 > Group 4. In the central fossa, the maximum discrepancy was also noted in Group 2 (CFD = 194.48 ± 13.71 μm). No significant differences were found in total cement space volume among the groups.

**Conclusion**. CAD/CAM crowns with grooves demonstrated clinically acceptable MG and IF values, with lithium disilicate crowns showing superior performance. These findings support the use of CAD/CAM technology for SCC restorations and underscore the importance of material selection and crown design for optimal clinical outcomes.

## INTRODUCTION

The occluso-cervical height is a crucial factor for ensuring the retention of dental crowns. Various factors can affect this occluso-cervical height, including extensive caries, fractures, occlusal discrepancies, developmental anomalies, and abfraction lesions. Additionally, morphological tooth wear resulting from attrition, abrasion, and erosion further complicates the situation. Achieving optimal retention and resistance forms in these conditions presents a significant challenge. Previous research indicates that the loss of crown retention is a primary factor contributing to the failure of conventional crowns and fixed partial prostheses (*Al Refai & Saker, 2018*). The dislodgment of the crown may result in functional and aesthetic difficulties. It was reported earlier that about 2/3rd of clinical cases failed due to short clinical abutments, resulting in crown dislodgment (*Sayed et al., 2024*).

It is recommended to have four mm as the minimal occluso-cervical height for preparations on the posterior teeth, but in cases of short clinical crowns/abutment (SCC), this is extremely difficult to achieve, and it negatively impacts retention, causing serious problems (*Sayed et al., 2024*; *Sharma et al., 2012*; *Anoop, 2018*). In clinical practice, these cases are addressed through either periodontal surgical procedures before crown preparation, which increases both the cost and complexity of treatment, or through the use of auxiliary retentive features (ARF) such as grooves, boxes, pins, and ledges to enhance the retention of the restoration. Grooves are the most commonly used option due to their conservative approach and ease of preparation. Similar to conventional crowns, crowns placed on short abutments must maintain consistency in marginal gap and internal fit (MG and IF) to ensure long-term clinical success. Any discrepancies in MG and IF can lead to issues such as plaque accumulation, caries, or periodontal disease, particularly with subgingival margins, which may result in cement dissolution, bacterial infiltration, and decreased fracture resistance (*Rizonaki et al., 2022*; *Contrepois et al., 2013*). *Holmes et al. (1989)* established a framework for categorizing the marginal gap (MG) and internal fit (IF) of dental crowns. According to the specifications outlined by the American Dental Association (ADA) in specification no. 8, the recommended film thickness for luting agents ranges from 25 µm to 40 µm (*Rizonaki et al., 2022*; *ADA, 1974*). However, achieving restoration precision within this specified range poses clinical challenges, as marginal gaps and cement thicknesses extending to 120 µm and 150 µm have been recognized as clinically acceptable limits (*Rizonaki et al., 2022*).

Recent trends in dentistry indicate a growing preference for metal-free materials used in posterior restorations. All-ceramic restorations are increasingly in demand due to their superior aesthetic appeal, compatibility with biological tissues, and ability to withstand compressive forces during chewing (*Alsubaiy et al., 2021*; *Chaturvedi et al., 2021*). With the development of computer-aided design and computer-aided manufacturing (CAD-CAM) techniques and ceramic materials, all ceramic restorations have proven to be the treatment of choice in many cases, including short clinical abutment cases. Like any other short crown cases, these CAD-CAM made all ceramic crowns also requires auxiliary retentive features. However, discrepancies may arise in these fabricated restorations due to constraints in the software's design capabilities, accompanied by hardware restrictions related to 3D image acquisition, and the intricacies of the milling procedure (*Hamza & Sherif, 2017*).

Studies had been conducted evaluating the marginal gap and internal fit of complete all ceramic crowns. MG-IF was reported as mean and median 28 μm and 74 μm respectively for e.max CAD crowns (*Hamza & Sherif, 2017*; *Miwa et al., 2016*; *Ng, Ruse & Wyatt, 2014*; *Hamza et al., 2013*). *Hamza & Sherif (2017)* concluded the marginal gap as 28.1 ± 7.9 μm and 40.2 ± 6.7 μm for e.max CAD crowns made by two CAD-CAM systems. *Mously et al. (2014)* determined median marginal gap as 49.35 μm, 46.65 μm and 55.18 μm for marginal gap values of e.max CAD crowns with 30-, 60-, and 100-mm die spacer thickness. The values obtained from the e.max Press group were found to be considerably lower in comparison to those recorded in the e.max CAD groups, with a statistically significant difference ($P = .005$). It is recommended to exercise caution when calculating the mean marginal gap (MG) values following the cementation process, as these post-cementation values were noted to be greater than those measured before cementation, irrespective of the specific crown material utilized or the finish line design (*Mounajjed et al., 2018*; *Peroz et al., 2019*; *Aktaş et al., 2025*; *Şentürk et al., 2025*; *Yenidünya, Misilli & Ocak, 2024*).

The complete sitting of a crown may be influenced by various factors including the viscosity of the cement and the cementation method. Still, along with this the dimensional variations in the cement space may also be influenced by the method of setting of resin cement (*Rizonaki et al., 2022*; *Alharbi et al., 2018*). The cement film filling the internal void must be uniform and minimal in thickness. When the thickness of the cement layer surpasses the suggested limits, it typically results in increased water absorption, which can ultimately cause the deterioration of the cement structure (*Rizonaki et al., 2022*).

The MG and IF may also be affected by the auxiliary retentive features. *Tjan Sarkissian & Miller (1981)* demonstrated that the existence of grooves can influence the fit of the casting, complicating the seating process, with the complexity escalating alongside the number of grooves present. Nevertheless, it was recommended to utilize grooves to improve retention, as there was a significant enhancement in retention, when the cast crown extended into the groove instead of being obstructed. This represents the importance of auxiliary retentive features in SCC cases (*Sayed et al., 2024*).

Assessment of MG and IF of crowns can be done by various techniques which includes stereomicroscope for viewing it directly (*Keshvad et al., 2012*), scanning electron microscopy (*Chaturvedi et al., 2020*; *Rajput et al., 2022*), or optical microscope (*Al-Dwairi et al., 2019*); the technique involving cross-sectioning (*Contrepois et al., 2013*);

radiographic techniques (*Liedke et al., 2015*); the technique involving impression replica (*Rahme et al., 2008*); profilometry (*Mitchell, Pintado & Douglas, 2001*); digital method of quantitative evaluation (*Chaturvedi et al., 2020*; *Rajput et al., 2022*); and the most recent one is microcomputed X-ray tomography (μCT) (*Chaturvedi et al., 2020*; *Daou et al., 2018*; *Ersöz et al., 2024*; *Tamam et al., 2023*).

To the best of authors' knowledge, assessment of MG and IF in All ceramic crowns with auxiliary features using micro-ct (μCT), is not documented until now. There is a gap in literature describing this critical aspect of short clinical crowns. Thus, the present study was under taken to determine the marginal and internal fit and cement volume of different CAD/CAM ceramic crowns (zirconia and lithium disilicate) prepared with retentive auxiliary feature (groove), on short clinical crowns, by using micro-CT. The null hypothesis formulated was that there will be no effect on the marginal gap and internal fit or the cement volume of different CAD/CAM ceramic crowns with and without grooves.

## MATERIALS AND METHODS

The present study was conducted in King Khalid University, Abha, KSA, in Department of Prosthodontics, College of Dentistry and was approved with ethical waiver certificate by the institute's ethical committee (IRB/KKUCOD/ETH-W/2023-24/004). The study was performed in following steps as in flow chart (Fig. 1). Mandibular first molar #37 was prepared for all ceramic crown without groove and later in same tooth buccal retentive groove was made.

### Preparation parameters

The preparation parameters were chosen to resemble a short crown (the axial walls taper more than 6°, axial height three mm, anatomical occlusal diameter six mm, Axial reduction: 1.5 mm, Planar occlusal reduction of 1.5–2 mm, buccal functional cusp bevel and the finish line was a deep chamfer with one mm in width) to simulate the indication for auxiliary feature a retentive groove. The retentive groove was prepared (1 × 2 mm) on the mid buccal side 0.5 mm above the finish line and occlusally diverged. Groove was placed parallel to path of insertion (Fig. 2).

### Master metal die (MMD) fabrication

Initially the prepared tooth #37, without groove was scanned using laboratory desktop scanner (Ceramill Map 400; AmannGirrbach) to fabricate metal master die. The scanned file of the prepared tooth without groove was stored as standard tessellation language (STL) format and then transferred to CAD-CAM software (Cerec inLab 4.2; Dentsply Sirona). The MMD was fabricated using a 5-axis milling machine (Ceramill Motion 2; Amann Girrbach) and metal disc (cobalt-chromium (Wirobond C; BEGO GmbH) with 0.6 mm bur diameter, one mm and three mm following manufacturer's recommendation. After MMD was fabricated without groove, a groove was made on the mid buccal side of the previously prepared typodont tooth. After groove preparation similar process was followed for the fabrication of MMD with groove. Thus, total two MMD were made one with groove
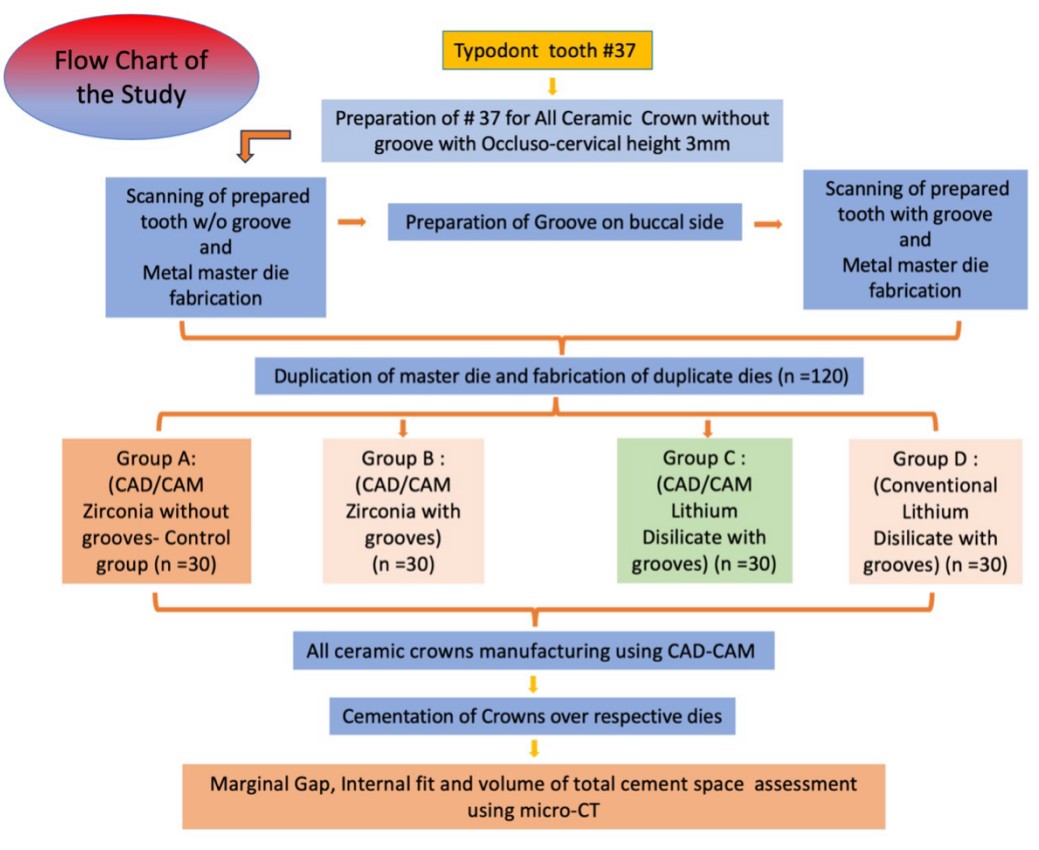

**Figure 1  Flow chart of study steps.**

and one without groove. Both the MMDs were duplicated to create working models using Type IV die stone (GC Fujirock EP; GC Europe).

## Working model fabrication and group division

Two prepared MMDs were used for duplicated to form working dies using Type IV die stone (GC Fujirock EP; GC Europe). To facilitate the duplication process, a specialized tray was designed, incorporating vent holes to enable the proper flow of excess material during the duplication with polyvinyl siloxane. In total, 120 dies were created, of which 30 were produced without grooves and 90 included grooves. The 30 dies without groove were allotted to control group while the grooved dies were randomly divided into three experimental groups and coded. Upon completion of the pouring process, all 120 dies were retrieved and meticulously examined for bubbles, voids, or any other defects. Any identified defective dies were discarded, prompting the need to pour new dies to ensure quality. The sample size 120 divided into four groups.

Group 1: (CAD/CAM Zirconia without grooves- Control group) ($n = 30$).
Group 2: (CAD/CAM Zirconia with grooves) ($n = 30$).
Group 3: (CAD/CAM Lithium Disilicate with grooves) ($n = 30$).
Group 4: (Conventional Lithium Disilicate with grooves) ($n = 30$).

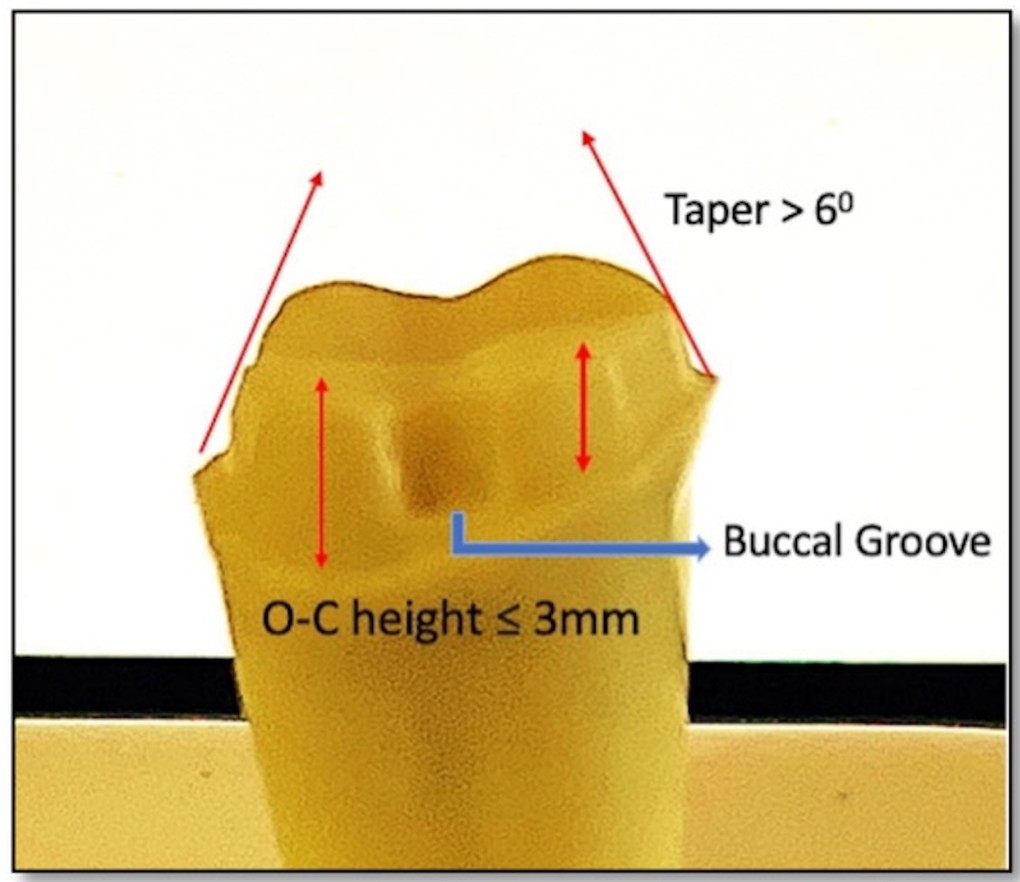

**Figure 2** Typodont tooth preparation representing short clinical crown of the abutment.

## CAD/CAM All ceramic crowns manufacturing procedure

All 120 dies were coded and scanned individually using desktop scanner (Ceramill Map 400; AmannGirrbach) and stl files were transferred into CAD-CAM software (Cerec inLab 4.2; Dentsply Sirona) (Fig. 3). The designing of the all ceramic crowns were done individually on the scanned working die with 20 μm cement space and after visual verification by chief researcher the milling was done using a 5-axis milling machine (Ceramill Motion 2; Amann Girrbach) and corresponding ceramic discs as per group division, following manufacturer's recommendation. For group A and B the Zr disc were used and for group C the partially crystalized lithium disilicate CAD/CAM blocks (IPS e.max CAD LT, Ivoclar Vivadent) were used. All-ceramic crowns, after the milling process, were placed in a ceramic furnace to achieve final temperatures of 1150 °C for zirconia (Zr) and 850 °C for lithium-disilicate (LiDS), with a duration of approximately 25 min to ensure complete crystallization. Each type of crown was then glazed with the appropriate glaze formulated for the respective ceramic material. CAD-CAM milling process was selected due to its robust accuracy and repeatability. One of the primary benefits of CAD-CAM milling is its ability to produce restorations with exceptionally low marginal gaps. Recent studies have found
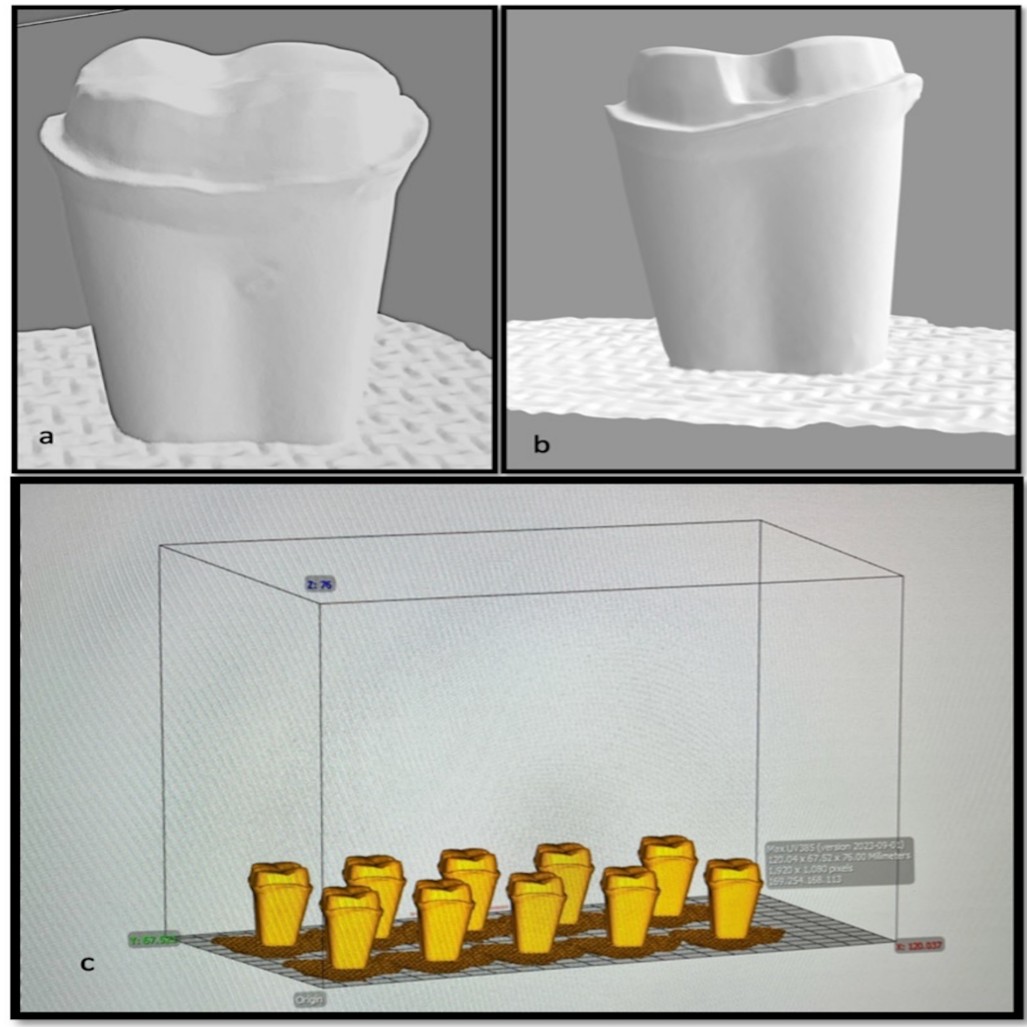

**Figure 3** Snapshot of stl.file of prepared tooth (A) without groove; (B) with groove; (C) dies arranged on CAD software.

that milled crowns typically exhibit marginal fit measurements that are acceptable, often reported in ranges around 30 to 50 µm (*Mounajjed, Layton & Azar, 2016*; *Alqahtani et al., 2025a*). This level of precision is crucial for maintaining periodontal health and ensuring longevity of restorations, as evidenced by studies documenting the superior marginal integrity of CAD-CAM fabricated crowns compared to those produced by 3D printing methods (*Al-Halabi et al., 2021*; *Mohan Kumar et al., 2024*). Furthermore, CAD-CAM milling eliminates the variability introduced by the manual processes associated with traditional crown fabrication, thereby contributing to consistency in quality and patient outcomes (*Fard et al., 2019*).

Another noteworthy benefit of CAD-CAM milling is its contribution to material efficiency. Although initial material waste in milling can be higher compared to additive manufacturing, advancements in milling technology have led to a reduction in wear on
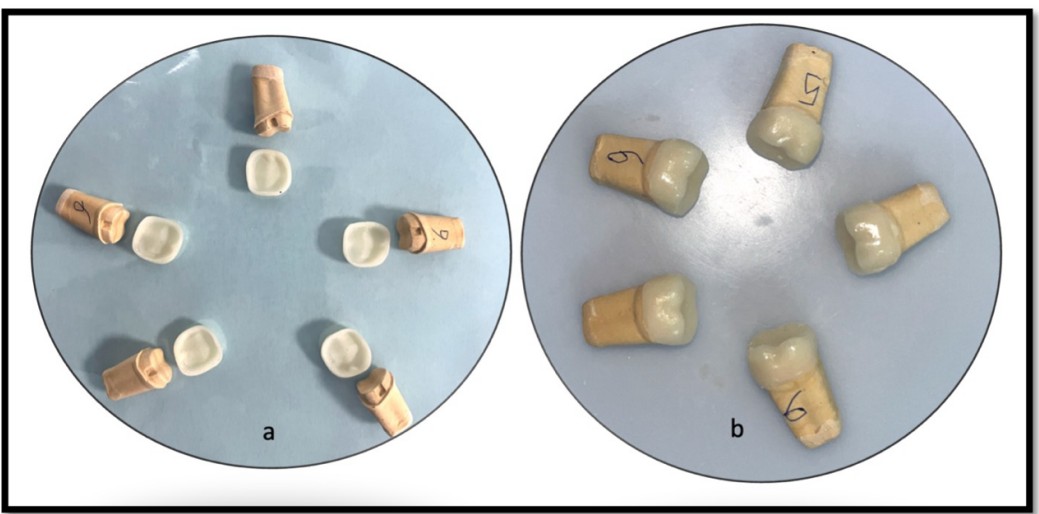

**Figure 4** Representative image showing (A) Milled crowns with groove; (B) cemented crowns over respective dies.

milling tools and materials, which are now more economically viable for dental practices due to the ability to produce restorations in-office, reducing dependence on external laboratories (*Al Hamad et al., 2022*).

CAD-CAM technology also facilitates the use of diverse materials that enhance the mechanical properties of crowns, such as lithium disilicate and zirconia, known for their strength and aesthetic qualities. Research has shown that crowns made from these materials demonstrate superior fracture resistance when produced *via* CAD-CAM milling, allowing for thinner designs without compromising structural integrity (*Oliveira et al., 2024*; *Alqahtani et al., 2025b*). In contrast, while 3D printing methods can reduce material waste and offer more complex internal structures, they sometimes struggle to match the mechanical strength and marginal fit accuracy of milled crowns, leading to concerns regarding their long-term clinical performance (*Al-Halabi et al., 2021*; *Assiri et al., 2025*).

## Cementation procedures

The crowns under examination were affixed utilizing a self-etching, self-adhesive resin cement, specifically Maxcem Elite (Kerr, CA, USA), in accordance with the manufacturer's guidelines. The automixed cement was evenly applied to the internal surface of the ceramic crown, which was then meticulously positioned onto the designated coded die with a finger pressure of 50 N. Any excess cement was subsequently removed with micro brushes. Light curing was performed for 30 s on all restorative surfaces, including mesial, distal, lingual, and buccal aspects, using a UV-polymerization lamp to finalise the bonding process. A trained technician blinded from study carried out all laboratory procedures under guidance of chief researcher and cementation procedure was carried out by chief researcher. (Fig. 4)
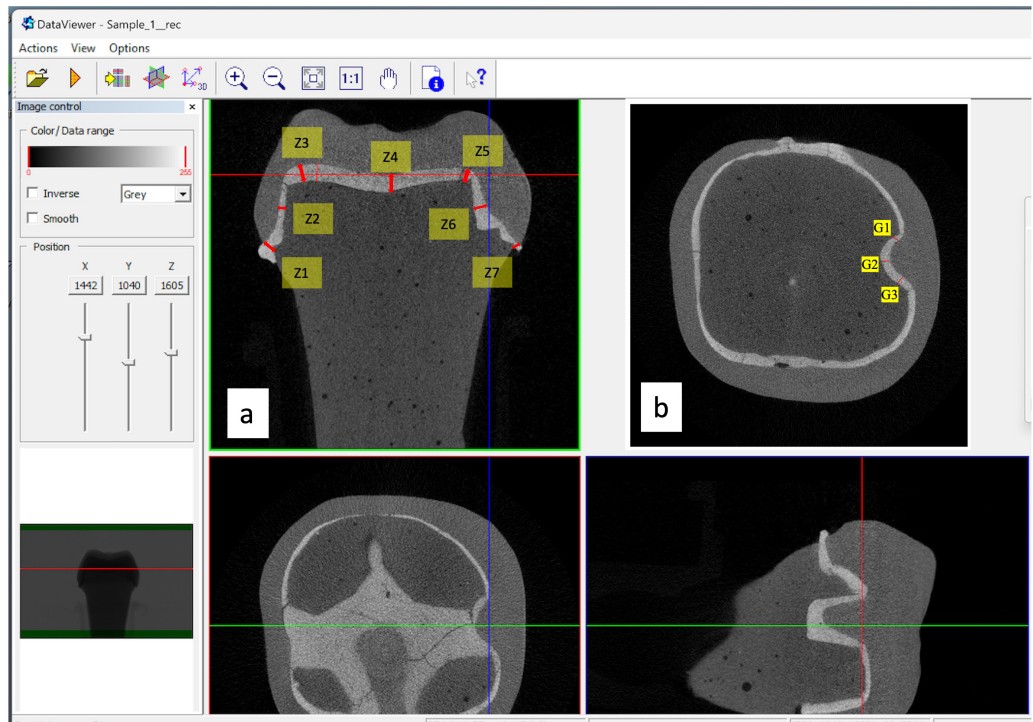

**Figure 5** **Micro-CT images showing seven zones and three groove area of measurement.**

## Marginal gap and internal fit assessment

The MG-IF of different CAD/CAM ceramic crowns (zirconia and lithium disilicate) prepared with and without retentive features (grooves) were assessed by using micro-CT. The crowns cemented over the study models were placed in the micro-CT machine for measurements. The measurements of the gaps were performed in two planes, at seven zones (Z) each (Z1 to Z7). The evaluation of crown fit concerning marginal and internal discrepancies is crucial for ensuring optimal clinical outcomes. In measuring these discrepancies, the gaps were categorised into four positions: absolute marginal discrepancy (AMD), axial wall discrepancy (AWD), cuspal area discrepancy (CAD), and central fossa discrepancy (CFD). Each category provides insight into the adaptation of restorative crowns, which can influence their long-term effectiveness and patient satisfaction (Figs. 5A and 5B).

AMD pertains to the gap measurement at the crown margin. This gap is between the farthest edge of the restoration's margin and the preparation's outer marginal line. Studies indicate that AMD is a critical factor in assessing the overall accuracy of the restoration as it relates directly to potential microleakage and the risk of secondary caries. *Chaturvedi et al. (2020)* emphasizes that systematic measurement of these marginal discrepancies is vital for evaluating restoration integrity and predicting clinical longevity. The zone 1 and 7 at margins were considered for measurement to determine the AMD (Fig. 5A).

To investigate internal discrepancies, perpendicular distances from the crown's inner surface to the preparation's outer surface were measured. AWD was assessed and refers
to the gap between the crown's internal surface and the prepared tooth's axial walls. In Fig. 5A, Zone 2 and Zone 6 represents the AWD. Studies reported that discrepancies in this area can significantly affect the retention and stability of the crown (*Aktaş et al., 2025*; *Chaturvedi et al., 2020*; *Alqahtani et al., 2025a*). *Alqahtani et al. (2025a)* report that poorly adapted axial walls may lead to an increased potential for crown failure under functional load due to inadequate cement retention.

CAD and CFD assess the crown's fit in specific occlusal regions. CAD examines the areas where cusps meet, while CFD targets fit measurements in the central region of the crown occlusal surface. Previous work indicated that cuspal areas often present the highest incidence of misfit, primarily attributed to the limitations of milling tool designs and techniques that fail to capture intricate anatomical contours adequately (*Zimmermann et al., 2019*). Zone 3 and Zone 5 represent CAD and CFD by zone 4 (Fig. 5A). These zones are crucial in predicting the restoration's fracture resistance, as occlusal discrepancies may lead to biomechanical failures under masticatory forces (*Oguz et al., 2021*).

In the auxiliary retentive groove area (G) measurements were performed at mesial, distal and floor wall. This selection was strategically based on previous studies, highlighting these areas' significance in maintaining crown retention and overall restoration integrity (*Pasha, Saleem & Bilal, 2023*; *Alsubaiy, 2023*). Each zone was chosen due to its anatomical relevance and potential behavioral reliance of the crown, overall retention of the crowns, ensuring that the regions assessed were clinically relevant and reflective of areas most likely to influence fit and performance.

In each zone, points were selected and measured two times by primary investigator (calibrated for micro-ct use) and trained technician of micro-ct, at an interval of 4 days and average was taken as final reading (Fig. 5B). Total 40 points were measured for each crown (seven in each coronal section; three in sagittal section in groove area per operator each time (8 marginal + 8 axial + 12 occlusal + 12 sagittal groove - by two operators twice), thus 1,200 readings were taken in each group ($n = 30$). This dual approach was designed to minimize individual measurement variability and reinforce our data collection's reliability. Each measurement point was evaluated twice, with a minimum interval of four days between assessments, allowing for any potential variations in technique or equipment to average out. This procedural rigor ensures replicability of our findings, aligning with recommendations from literature concerning twin assessments to enhance precision in measurements (*Tsai et al., 2020*). All data was systematically collected and evaluated utilizing statistical analysis software (IBM SPSS Statistics for Windows v22; IBM Corp). The volumetric assessments of the total cement volume (TCV) alongside porosity measurements exposed to the external environment (VP) were executed directly using Definiens.

## Micro-CT analysis

A comprehensive analysis of 120 specimens for marginal and internal fit was conducted utilizing a quantitative microcomputed tomography scanner (Skyscan 1275; Bruker micro-CT, Kontich, Belgium). The evaluation focused specifically on examining the marginal and internal fit of each specimen. Eighteen designated measuring sites, comprising nine per section, were meticulously chosen to assess cement thickness. The scanning process was

executed at 80 kVp and 125 mA, leveraging a pixel resolution of 24 µm, with a 0.4-degree rotational step, a three-frame average for enhanced accuracy, combined with a 1-mm aluminum filter to mitigate potential artifacts.

Artifact removal and image enhancement were conducted using NRecon software v.1.6.4.8 (Bruker micro-CT), which facilitated corrections for ring artifacts, smoothing, beam hardening, and post-alignment adjustments. Individual reconstruction of each scanned specimen was performed, yielding axial cross-sections *via* Data Viewer v.1.5.6.2 (Bruker micro-CT). For standardization, the screw channel in each die was oriented perpendicularly across all three axes, and imaging was carried out in both midcoronal and midsagittal planes. Each generated image underwent processing with CTAn software v.1.18.4.0 (Bruker micro-CT) to quantitatively determine the linear gap between the crown and corresponding die.

In preparation for the analysis of the ceramic crowns, a series of micro-CT scans followed strict calibration procedures to optimize image quality and contrast. A resolution of four µm per pixel was employed for each ceramic specimen, producing data reconstructed with a voxel resolution of 10 µm. The resultant reconstructed files (CTR) were then converted to tagged image file format (TIFF) for subsequent segmentation and measurement processes. All measurements were recorded in micrometers (µm) and were distributed throughout the entire preparation to accurately assess the marginal and internal fit of the ceramic crowns. Finally, volume rendering was performed using CT vox software (Bruker micro-CT, version 3.3.0 r1403) to calculate the total cement space volume (VTCS) in cubic millimeters ($mm^3$) (Fig. 5).

### Statistical analysis

The statistical examination was conducted employing SPSS Statistics (IBM Corporation, Armonk, USA, version 21), adhering to a significance threshold of 95%. Various descriptive statistics were derived, including the measured variables' mean, minimum, maximum, and standard deviation. The findings are illustrated through vertical bar charts that incorporate the standard deviations for clarity. The normality of the data was assessed by using the Shapiro–Wilk test and homogeneity of variances through Levene's test. To analyse the differences between the groups, a one-way ANOVA was utilized, followed by a *post hoc* Tukey test to further explore the results.

## RESULTS

The results of the study showed significant differences in the marginal and internal fit of the control group (CAD-CAM Zr-without groove) and other groups (Gr-2,3,4 with groove) ($P << 0.001$). The distribution of the outcome variable (retentive forces) by Shapiro Wilk normality test showed *p*-value > 0.05 (0.28). Group 4 had maximum AMD and AWD ((AMD = Z−1 = 197.36 ± 10.56; Z−7 = 226.5 ± 8.24); (AWD = Z−2 = 150.05 ± 10.89; Z−6 = 169.38 ± 10.57)) followed by Gr 2 > Gr 3 > Gr1. The discrepancy at the cuspal area was maximum in Gr-2 (CAD (Z−3 = 194.94 ± 16.28; Z−5 = 197.41 ± 10.36)) followed by Gr-3 > Gr-1 > Gr-4. While in the central fossa *i.e.,* Zone 4 the maximum discrepancy

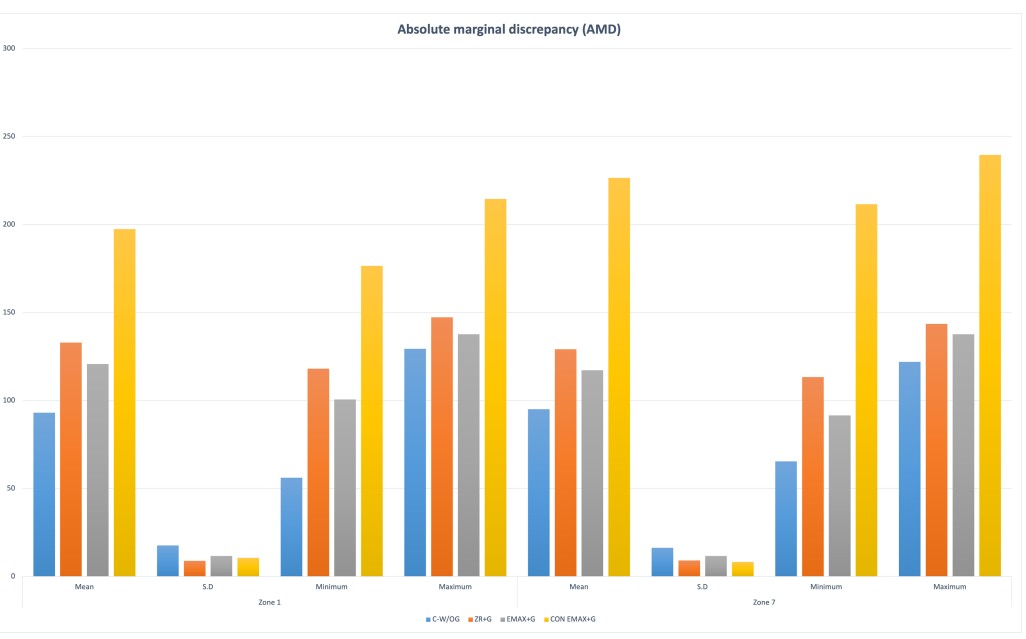

**Figure 6 Groupwise absolute marginal discrepancy (AMD) at Zone 1 and Zone 7.**

was noted in Gr-2 (CFD = 194.48 ± 13.71) and minimum in Gr-1(CFD = 124.44 ± 11.73) (Figs. 6, 7, 8).

The *post hoc* Tukey HSD Test (*p*-value < 0.001) was used for multiple comparison between the groups, and it confirmed the presence of statistically significant differences in the measurements of marginal and internal fit across these groups.

The internal fit at groove area was assessed separately at the three points (G-1,G-2,G-3). The maximum discrepancy was found in Gr-4 followed by Gr-3 and Gr-2. At point G2 the maximum gap (244.09 ± 10.21) was noted. The statistically significant difference (*P* < 0.001) was determined for intergroup comparison with *post hoc* Tukey HSD Test. (Tables 1, 2 and Fig. 9).

The assessment of volume of total cement space (VTCS) (mm³) revealed that Gr-4 had maximum cement volume space 22.79 ± 2.21 Gr-1 had minimum cement volume space 20.969 ± 3.25. All pairwise comparisons between the groups were not significant, with *p*-values greater than 0.05 for all comparisons (Tables 3 and 4).

Overall marginal and internal fit was influenced by presence or absence of groove, material used and the technique of fabrication *i.e.,* digital or conventional. The least marginal gap was noted in crowns without groove (Gr-1) compared to crowns with groove. It was minimum in in CAD-CAM (digitally made) Emax-crown (Gr-3) compared to CAD-CAM zirconia crowns (Gr-2) and conventionally made Emax crowns (Gr-4).

## DISCUSSION

Short clinical crowns impose a challenging task for the clinician in terms of achieving adequate retention for full coverage crowns. During function the vector of forces are
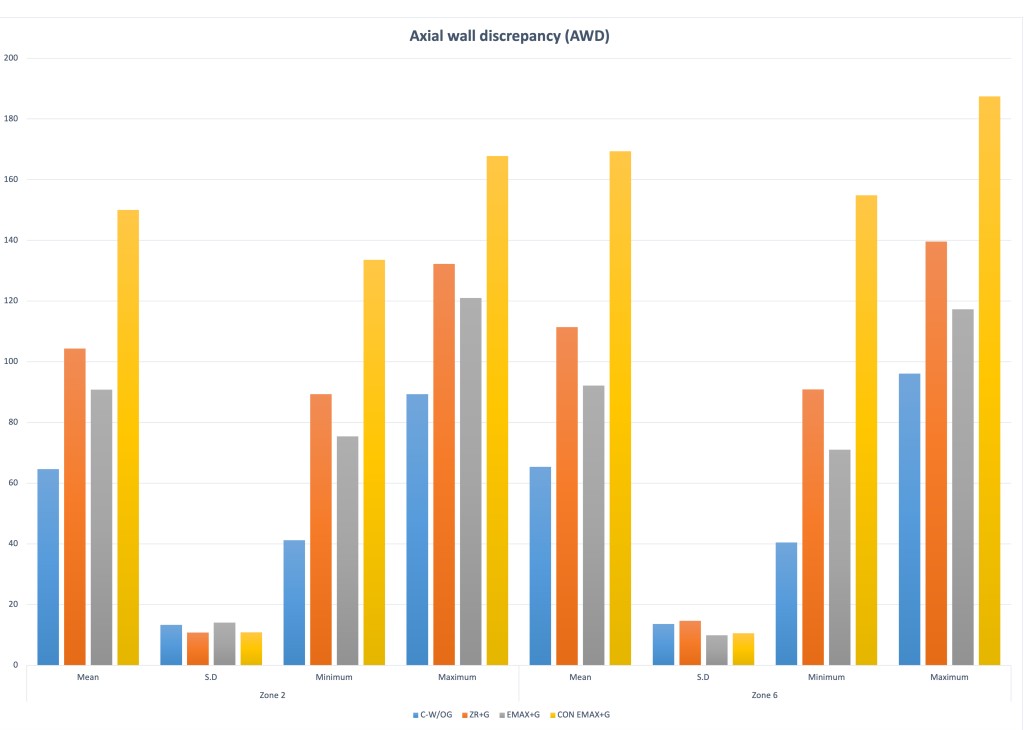

**Figure 7** Groupwise axial wall discrepancy (AWD) at Zone 2 and Zone 6.

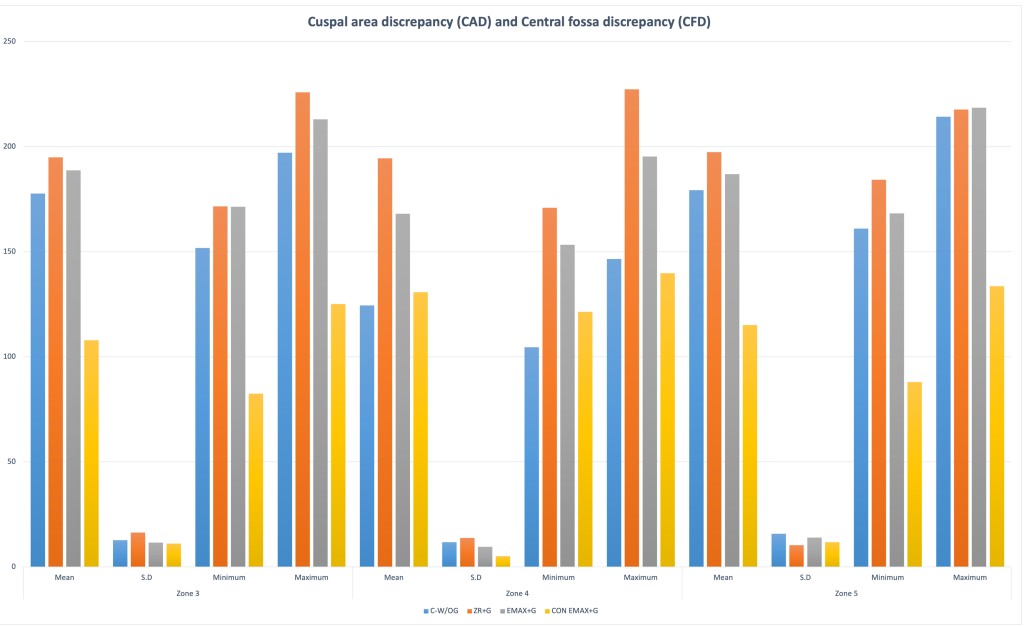

**Figure 8** Groupwise cuspal area discrepancy (CAD) at Zone 3 and Zone 5 and central fossa discrepancy (CFD) at Zone 5.

**Table 1  Zone wise multiple comparison between the groups using *post hoc* Tukey HSD test.**

Multiple comparisons (Tukey HSD)

| Dependent variable | | Mean difference - absolute marginal discrepancy (AMD) | | Mean difference - Axial wall discrepancy (AWD) | | Mean difference - Cuspal area discrepancy (CAD) | | Mean difference - Central fossa discrepancy (CFD) | Sig. | Result |
|---|---|---|---|---|---|---|---|---|---|---|
| | | [at Zone 1- Std. Error = 3.25690] | [at Zone -7 - Std. Error = 3.03950] | [at Zone-2 - Std. Error = 3.18934] | [at Zone-6 - Std. Error = 3.19174] | [at Zone-3 - Std. Error = 3.37451] | [at Zone-5 - Std. Error = 3.38912] | [at Zone-4 - Std. Error = 2.72142] | | |
| C-W/OG | ZR+G | −39.86203* | −34.03053* | −39.69100* | −46.06967* | −17.23667* | −18.15267* | −70.04033* | <0.001 | *Sig |
| | EMAX+G | −27.60470* | −22.11387* | −26.18367* | −26.82133* | −10.98567* | −7.667 | −43.62967* | <0.001 | *Sig |
| | CON EMAX+G | −104.27257* | −131.38953* | −85.42000* | −104.01100* | 69.83400* | 64.14000* | −6.25233 | <0.001 | *Sig |
| ZR+G | C-W/OG | 39.86203* | 34.03053* | 39.69100* | 46.06967* | 17.23667* | 18.15267* | 70.04033* | <0.001 | *Sig |
| | EMAX+G | 12.25733* | 11.91667* | 13.50733* | 19.24833* | 6.251 | 10.48567* | 26.41067* | <0.001 | *Sig |
| | CON EMAX+G | −64.41053* | −97.35900* | −45.72900* | −57.94133* | 87.07067* | 82.29267* | 63.78800* | <0.001 | *Sig |
| EMAX+G | C-W/OG | 27.60470* | 22.11387* | 26.18367* | 26.82133* | 10.98567* | 7.667 | 43.62967* | <0.001 | *Sig |
| | ZR+G | −12.25733* | −11.91667* | −13.50733* | −19.24833* | −6.251 | −10.48567* | −26.41067* | <0.001 | *Sig |
| | CON EMAX+G | −76.66787* | −109.27567* | −59.23633* | −77.18967* | 80.81967* | 71.80700* | 37.37733* | <0.001 | *Sig |
| CON EMAX+G | C-W/OG | 104.27257* | 131.38953* | 85.42000* | 104.01100* | −69.83400* | −64.14000* | 6.25233 | <0.001 | *Sig |
| | ZR+G | 64.41053* | 97.35900* | 45.72900* | 57.94133* | −87.07067* | −82.29267* | −63.78800* | <0.001 | *Sig |
| | EMAX+G | 76.66787* | 109.27567* | 59.23633* | 77.18967* | −80.81967* | −71.80700* | −37.37733* | <0.001 | *Sig |

Notes.
*Significant.

**Table 2  Groove area wise multiple comparison between the groups using *post hoc* Tukey HSD Test.**

Multiple comparisons (Tukey HSD)

| Dependent variable | | Mean difference (I-J) | Mean difference (I-J) | Mean difference (I-J) | Sig. | Result |
|---|---|---|---|---|---|---|
| | | [at G-1- Std. Error = 1.95711] | [at G-2 - Std. Error = 2.00425] | [at G-3 - Std. Error = 1.85921] | | |
| C-W/OG | ZR+G | −105.96833* | −187.16033* | −173.50233* | <0.001 | *Sig |
| | EMAX+G | −88.75967* | −148.63633* | −132.26233* | <0.001 | *Sig |
| | CON EMAX+G | −131.50400* | −244.09100* | −164.83567* | <0.001 | *Sig |
| ZR+G | C-W/OG | 105.96833* | 187.16033* | 173.50233* | <0.001 | *Sig |
| | EMAX+G | 17.20867* | 38.52400* | 41.24000* | <0.001 | *Sig |
| | CON EMAX+G | −25.53567* | −56.93067* | 8.66667* | <0.001 | *Sig |
| EMAX+G | C-W/OG | 88.75967* | 148.63633* | 132.26233* | <0.001 | *Sig |
| | ZR+G | −17.20867* | −38.52400* | −41.24000* | <0.001 | *Sig |
| | CON EMAX+G | −42.74433* | −95.45467* | −32.57333* | <0.001 | *Sig |
| CON EMAX+G | C-W/OG | 131.50400* | 244.09100* | 164.83567* | <0.001 | *Sig |
| | ZR+G | 25.53567* | 56.93067* | −8.66667* | <0.001 | *Sig |
| | EMAX+G | 42.74433* | 95.45467* | 32.57333* | <0.001 | *Sig |

Notes.
*Significant.

applied to the crowns which may result its dislodgment. Auxiliary retentive features are recommended to overcome this problem with grooves being most common. With the developing trends in aesthetics, patients are becoming more specific for all ceramic crowns even in posterior teeth with short clinical crowns (*Sayed et al., 2024*; *Sharma et al., 2012*; *Mously et al., 2014*; *AlShaarani et al., 2019*).

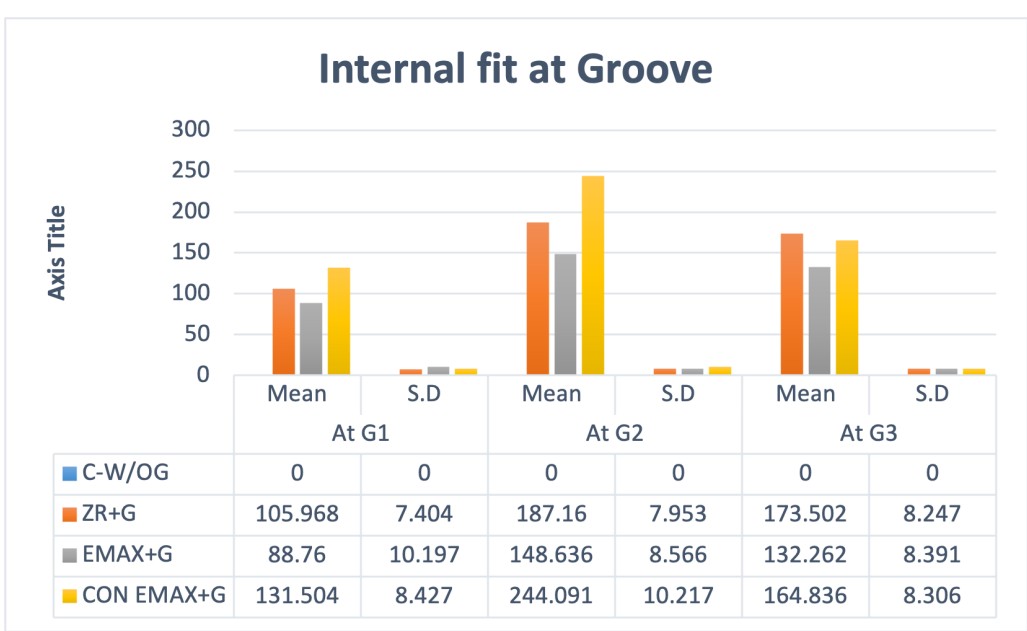

**Figure 9** Groupwise auxiliary retentive groove area (G) measurements at G1, G2, G3.

**Table 3 One-way ANOVA between the groups in terms of volume.**

| Descriptive stats | | Mean | S.D | N | Minimum | Maximum | *F*-Test | *P*-Value | Result |
|---|---|---|---|---|---|---|---|---|---|
| Vol in mm³ | C-W/OG | 20.969 | 3.257 | 30 | 16.32 | 28.23 | | | [**]Non-Sig |
| | ZR+G | 21.712 | 2.607 | 30 | 18.51 | 29.09 | 2.130 | 0.100 | |
| | EMAX+G | 21.917 | 3.067 | 30 | 17.92 | 29.67 | | | |
| | CON EMAX+G | 22.793 | 2.214 | 30 | 20.13 | 29.34 | | | |

**Notes.**
[**]Non-significant.

Numerous prior studies have demonstrated that CAD-CAM crowns exhibit superior accuracy compared to conventional crowns in full coverage restorations. However, despite CAD/CAM technology advancements, all-ceramic crowns are not devoid of complications. Issues such as inadequate internal fit, challenges related to adhesion, loss of marginal integrity, microleakage, and the potential for tooth hypersensitivity are prevalent when these crowns are evaluated against traditionally fabricated materials (*Papadiochou & Pissiotis, 2018*; *Aswal et al., 2023*). The studies on short crown with grooves are lacking thus present study was undertaken to assess the marginal and internal fit and volume of total cement space (VTCS) for different CAD/CAM (Zirconia and Lithium Disilicate) and conventional all ceramic crowns (lithium disilicate) with retentive auxiliary feature (groove), on short clinical abutment by using micro-CT. The study's results rejected the null hypothesis as the marginal and internal fit of CAD-CAM crowns were better than conventional crowns. Also, the CAD-CAM zirconia crowns without grooves had better marginal fit than CAD-CAM made zirconia and lithium disilicate crowns with groove and conventionally made lithium disilicate crowns with grooves. But the other null hypothesis

**Table 4  Multiple comparison between the groups using *post hoc* Tukey HSD test in terms of volume.**

| Multiple comparisons (Tukey HSD) | | | | | | |
|---|---|---|---|---|---|---|
| | Dependent variable | | Mean difference (I-J) | Std. Error | Sig. | Result |
| Vol in mm³ | C-W/OG | ZR+G | −0.74333 | 0.72703 | 0.737 | N-Sig |
| | | EMAX+G | −0.94833 | 0.72703 | 0.562 | N-Sig |
| | | CON EMAX+G | −1.82400 | 0.72703 | 0.064 | N-Sig |
| | ZR+G | C-W/OG | 0.74333 | 0.72703 | 0.737 | N-Sig |
| | | EMAX+G | −0.20500 | 0.72703 | 0.992 | N-Sig |
| | | CON EMAX+G | −1.08067 | 0.72703 | 0.449 | N-Sig |
| | EMAX+G | C-W/OG | 0.94833 | 0.72703 | 0.562 | N-Sig |
| | | ZR+G | 0.20500 | 0.72703 | 0.992 | N-Sig |
| | | CON EMAX+G | −0.87567 | 0.72703 | 0.625 | N-Sig |
| | CON EMAX+G | C-W/OG | 1.82400 | 0.72703 | 0.064 | N-Sig |
| | | ZR+G | 1.08067 | 0.72703 | 0.449 | N-Sig |
| | | EMAX+G | 0.87567 | 0.72703 | 0.625 | N-Sig |

for cement volume was accepted as there was no significant difference in the cement volume of different crowns with grooves.

The marginal gap and internal fit of dental restorations influence their clinical efficacy. An increased marginal gap results in a thicker layer of luting cement, which may dissolve in the oral environment. This dissolution can adversely affect the restorations' success and durability over time (*Vasiliu, Porojan & Porojan, 2020*; *Gaikwad et al., 2018*).

*Kaufman, Coelho & Colin (1961)* analyzed the impact of preparation height on retention or resistance form and established a linear correlation indicating that lower preparation heights result in reduced retention. The findings also demonstrated an enhancement in retention with the integration of retentive features as opposed to traditional preparations. Specifically, the inclusion of retentive grooves exhibited a statistically significant improvement in retention compared to the retention achieved through box preparations. Supporting these findings, *Vinaya et al. (2015)* similarly assessed proximal grooves against boxes and determined that grooves significantly augmented retention relative to boxes. Furthermore, *Haritha et al. (2020)* highlighted that adding a horizontal groove on each side of a prepared tooth notably increased crown retention. Accordingly, the present study considered grooves as an auxiliary feature in the evaluation of SCC.

The current investigation evaluated the characteristics of Zr and LiDS crowns. Zirconia (Zr) has been recognized as a credible treatment method for the restoration of individual teeth, as supported by prior research (*Mühlemann et al., 2020*; *Cagidiaco et al., 2019*; *Abualsaud & Alalawi, 2022*). This endorsement stems from Zr's intrinsic aesthetic appeal and robust mechanical attributes (*Mühlemann et al., 2020*; *Abualsaud & Alalawi, 2022*). The base material of Zr is available in two distinct forms: fully sintered, which is suited for hard milling, and partially sintered, which is appropriate for green stage milling. Each of these manufacturing processes presents specific drawbacks. Hard milling can result in issues such as crack development within the restoration, wear on milling tools, extended milling

durations, and variability in surface topographies (*Xu et al., 2022*). Conversely, green stage milling necessitates an increase in design dimensions to account for the reduction that occurs during the sintering process (*Papadiochou & Pissiotis, 2018*; *Aswal et al., 2023*). The resultant surface quality and configuration of the restoration are significantly affected by both the geometry and dimensions of the milling tools employed, as well as the number of milling axes available on the machinery used (*Alsubaiy et al., 2021*; *Chaturvedi et al., 2021*; *Chaturvedi et al., 2020*; *Mühlemann et al., 2020*; *Roy et al., 2021*). In this study, we opted for partially sintered Zr utilizing green stage milling, due to its advantages compared to fully sintered zirconia.

Although it is typically advised that LiDS crowns be utilized in the anterior region, the performance of CAD-CAM LiDS crowns demonstrates fracture resistance surpassing the maximum bite force documented for the first molar area following a fatigue assessment. This indicates that their clinical application can be deemed acceptable. Furthermore, lithium disilicate glass-ceramics (LDS) provide enhanced aesthetic properties (*Jurado et al., 2022*, *Spitznagel et al., 2024*), a reliable clinical long-term survival (*Spitznagel et al., 2024*), high clinical performance survival rates of 93% after 6, years (*Aziz et al., 2022*) and 80.1% after 15, years (*Rauch et al., 2023*). The CAD/CAM fabrication process for LDS reconstructions offers a consistent and uniform approach to manufacturing. This method enhances both the efficiency of time and the reduction of costs, leading to improved overall productivity in the development of these reconstructions (*Jurado et al., 2022*; *Spitznagel et al., 2024*; *Aziz et al., 2022*). Literature indicates that crowns crafted from lithium disilicate ceramics are regarded as some of the finest restorative materials available. This high regard is attributed to their excellent adhesive qualities, which facilitate a strong bond, as well as their ability to achieve micromechanical interlocking when used in conjunction with resin cements (*Jurado et al., 2022*; *Spitznagel et al., 2024*). *Gudugunta et al. (2019)* and *Dolev, Bitterman & Meirowitz (2019)* conducted an analysis to assess the marginal fit of hot-press *versus* CAD-CAM lithium disilicate crowns. The findings indicated that crowns produced using the CAD-CAM method demonstrated superior marginal fit compared to those formed by the pressing technique. Furthermore, *in vitro* studies have indicated that fully anatomical e.max CAD crowns possess a fracture resistance level that is appropriate for use in posterior, monolithic restorations. Additionally, these crowns exhibit greater resistance to fatigue under cyclic loading conditions compared to veneered zirconia, which has a higher susceptibility to chipping (*Zarone et al., 2019*). Therefore, in this study, the CAD-CAM method was utilized for the creation of crowns made from both zirconia and lithium disilicate materials.

The type of milling machine could affect the adaptability of the restoration, particularly if it has a complicated shape with auxiliary features like groove areas, and internal angles. A five-axis milling machine was employed to produce the restoration in the current investigation (*Zarone et al., 2019*; *Song, Ren & Yin, 2016*). The five-axis milling machines showed the best accuracy of fit when different milling units were tested (*Alajaji et al., 2017*). Additionally, the adaption of a restoration might be influenced by the bur size and form of the milling unit. For intricate shape machining, a minor diameter of 0.6 mm is recommended (*Kim et al., 2016*), same has been used in the present study.

Measuring the IF of the restoration with 2D analysis yields a limited set of measurement points. As a result, outcomes could not accurately reflect how well the restoration fits (*Keshvad et al., 2011*; *Nagi, Fouda & Bourauel, 2023*). Thus, the MG-IF of the restoration was assessed using 3D microcomputed tomography (*Nagi, Fouda & Bourauel, 2023*). Because it offers more point measurements than a 2D technique does, this technique has good validity and reliability, also helps in accessing the cement volume of the samples (*Nagi, Fouda & Bourauel, 2023*; *Boitelle et al., 2018*).

In the present study, the AMG is represented by Zone 1 and Zone 7. The mean MG of conventional LiDS crowns was highest followed by CAD-CAM Zr, CAD-CAM LiDS and least MG was recorded in control group CAD-CAM Zr without groove. Showing that digital fabrication method had provided better marginal integrity and less MG. This result was in association with the results of the *Berejuk et al. (2014)*. In their findings, they noted that the marginal fit was significantly worse in the conventional workflow, measuring 11.56 mm and 8.74 mm, whereas in the digital workflow, it was notably better, with measurements of 1.85 mm and 1.50 mm (*Lee, Son & Lee, 2020*). With the development of the Advanced machines the digital workflow has been reported to produce better marginal fit of prostheses than that of those fabricated using the conventional method (*Lee et al., 2020*; *Rapone et al., 2020*). The findings of *Örtorp et al. (2011)* starkly contrast with those of our current study, highlighting a significant issue. They observed a markedly inferior marginal fit in the digital workflow, measuring 222.5 mm and 124.6 mm, when compared to the traditional workflow. This discrepancy underscores the importance of re-evaluating the effectiveness of digital methods in comparison to conventional practices (118 $\mu$m; 49.7 $\mu$m).

The milling technique involves the creation of an object through a subtractive approach, utilizing cutting burs for material removal. In this context, the dimensions of the bur and the extent of its cutting trajectory emerge as critical constraints on the fabrication process. Although a bur with a diameter of 0.6 mm was employed in this study, it is noteworthy that the presence of sharp edges or an uneven surface can affect the overall quality of the fit. Consequently, an object with a smooth surface is likely to achieve a superior fit compared to its rougher counterpart (*Abualsaud & Alalawi, 2022*; *Aziz et al., 2022*).

On comparison of CAD-CAM LiDS and Zr crowns, the LiDS showed better results and there was statistically significant difference between them. This can be attributed to the structural resistance of Zr. The utilization of soft milled pre-sintered zirconia for crown production presents an essential consideration regarding its intrinsic limitation, specifically the necessity for a 25% oversizing of the framework prior to milling. This requirement arises because the material undergoes linear shrinkage after the sintering process, leading to dimensional inaccuracies. Such inaccuracies are particularly pronounced when dealing with complex geometrical frameworks, despite the simplification of milling procedures they afford. In contrast, LiDS crowns are fabricated from pre-crystallized blocks. The milling process occurs in this pre-crystallized state, and subsequent to milling, a thermal cycling process—conducted at temperatures between 840 °C and 850 °C for a duration of ten minutes—transforms metasilicate crystals into lithium disilicate at a rate of approximately 70%. This transformation enhances the material's flexural strength and toughness while

minimally impacting its overall dimensions (*Dolev, Bitterman & Meirowitz, 2019*; *Zarone et al., 2019*; *Nagi, Fouda & Bourauel, 2023*).

Consistent with earlier findings, the occlusal gap exhibited a greater size compared to both the axial and marginal gaps across all examined groups. Notably, the digital technique yielded the smallest measurements for the axial gaps (*Shamseddine et al., 2016*; *Bindl & Mörmann, 2005*; *Al Hamad et al., 2019*). The MG was least in control group followed by CAD-CAM crowns with groove of LiDS and Zr and LiDS conventionally made crowns. On the contrary the occlusal gap was maximum in CAD-CAM crowns with groove of Zr and LiDS followed by control group and least in conventionally made LiDS crowns with groove. On the other hand the axial gap was reversed from occlusal gap. These discrepancies would be associate with the grooves which resulted in more complex morphology to mill resulting in increased gaps in crowns with grooves compared to the control group with groove. These findings are in agreement with those in previous studies (*Colpani, Borba & Della Bona, 2013*), stating limitations of milling techniques in terms of burs and scanning in complex morphology compared to flat surfaces. This phenomenon can also be elucidated through the concept of hydraulic pressure generated by the cement. This pressure exerts a force on the material within the occlusal and marginal zones, a consequence of the convergence of the walls of the preparation. This design facilitates the movement of the cement material, enabling its ongoing displacement until it achieves the thinnest feasible layer (*Rudolph, Luthardt & Walter, 2007*). In spite of standardizing the cement gap 20 $\mu$m for all crowns this cement space affects the internal adaptation. The disparity in cement application may have been affected by several factors, including the dimensions and configuration of the milling tools, the manner of preparation, the quality of data acquisition, and the techniques employed in processing the digital information.

Numerous studies documented in the literature have assessed the marginal gap for cemented restorations, specifically identifying measurements between 100 to 200 mm and an internal gap ranging from 200 to 300 mm (*Svanborg, 2020*). These measurements are deemed clinically acceptable. The results obtained in this study fall within these established parameters.

The analysis volume of total cement space (VTCS) of revealed no substantial differences among the various groups, irrespective of the parameters set within the design software. This finding aligns with conclusions drawn from earlier studies (*Hamza et al., 2013*; *Mously et al., 2014*; *Mitchell, Pintado & Douglas, 2001*). It is essential to maintain a uniformly distributed cement space, as this is critical for ensuring optimal mechanical properties and effective retention of the restoration (*Al-Dwairi et al., 2019*; *Liedke et al., 2015*). In the current study, a similar observation emerged, with the mean VTCS ranging from $20.96 \pm 3.25$ to $22.79 \pm 2.21$.

The current research concentrated on a fully digital workflow, specifically examining milled crowns. Evidence suggests that CAD-CAM crowns featuring grooves demonstrate superior marginal gaps and interfacial fit compared to traditionally fabricated crowns. Consequently, based on the findings of this study, it is advisable for clinicians to consider utilizing CAD-CAM crowns with grooves in situations involving short clinical abutments.

## Limitations

The current study presents several limitations, with particular emphasis on the *in vitro* design, which does not fully replicate the complexities of an *in vivo* environment encountered in actual clinical practice. A significant limitation stems from the inability of the *in vitro* study to simulate masticatory forces accurately. The dynamic and cyclic loading that occurs during normal chewing is critical for assessing the durability and stability of dental crowns. Laboratory conditions cannot replicate the various forces exerted by diverse functional movements, including lateral and protrusive jaw movements, which can significantly affect the performance of dental restorations over time. Furthermore, the aging processes of both the dental materials and the cement used for fixation are not adequately represented in an *in vitro* setting. *In vivo*, factors such as temperature fluctuations, hydration, salivary interaction, and biofilm formation play pivotal roles in material degradation and the longevity of cement bonds, which were not accounted for in this study. Understanding the impact of these long-term interactions is essential, as they can lead to variability in crown performance once placed in a patient's mouth. While the study was conducted under strictly regulated conditions, ensuring uniformity across all samples and employing consistent fitting parameters, variations in the measured parameters and inconsistencies can largely be attributed to the subjective factors introduced by the operator, the dental technician, and the specifics of the cementation procedure, which remains difficult to standardize completely. Additionally, the study faced constraints such as limited image resolution, inaccuracies in landmark positioning, variations in cementation techniques due to uncontrolled finger pressure, and the use of dual-polymerizing resin cement. Nonetheless, the findings indicate that the marginal gap of the crowns fell within clinically acceptable ranges, regardless of the material types and techniques employed or the presence of grooves.

## Study importance and clinical implications

Understanding the relationship between MG, IF, and the presence of auxiliary features is crucial in evaluating the long-term success of CAD-CAM-fabricated all-ceramic crowns. Research has shown that discrepancies in the fit can significantly hinder the functional efficacy of the crown and lead to aesthetic and hygienic challenges (*Al Hamad et al., 2019*). Notably, previously researchers had elucidated the technical definitions surrounding MG and IF, framing the assessment of these variables as essential for quality assurance in restorative dentistry (*Oliveira et al., 2024*). With advancements in CAD-CAM technology, the production of metal-free restorations has seen a marked increase due to their satisfactory aesthetics and compatibility with biological tissues (*El Hayek et al., 2024*). However, the interaction between CAD-CAM fabricated crowns and auxiliary retentive features is still under-explored, especially in the context of SCCs.

There is a literature gap regarding the MG and IF in these specific cases, which this study aims to address using micro-computed tomography (μCT) for precise measurements. The novelty of using μCT in this assessment enhanced our understanding of how CAD-CAM technology can be optimized for clinical success, ultimately contributing to improved treatment outcomes for patients with short clinical abutments. The findings of the study

provide support clinicians in making informed decisions regarding the fabrication and implementation of SCCs especially with zirconia in complex cases. The zirconia crowns proved to be beneficial in restorative dentistry due to their superior mechanical properties, aesthetic appeal, and compatibility with oral tissues, making them an ideal choice for posterior restorations, particularly in cases with challenges such as short clinical crowns (SCC), thus ensuring a higher success rate and improved patient satisfaction in line with enhanced demand of all ceramic work by patients.

### Future research recommendations

Based on the results of this study, several recommendations are proposed to enhance future research on CAD-CAM zirconia crowns, especially for use in posterior teeth. Long-term clinical studies with follow-up periods of five years or more are needed to understand the durability, failure rates better, and patient satisfaction associated with these crowns. Research should further explore the role of auxiliary retentive features like grooves in improving crown stability across different clinical situations and tooth types, as current data on this topic are limited. Comparing various brands and types of zirconia, as well as other ceramic materials like lithium disilicate, could help identify differences in strength, fit, and patient experience. Additionally, studies should quantify the amount of wear zirconia crowns cause on opposing teeth and link this to outcomes like sensitivity and enamel health. The impact of different cementation techniques on the crown's fit and strength also requires evaluation, considering how operator methods and cement type may affect results.

## CONCLUSION

The study provides significant insights into the marginal and internal fit of CAD/CAM all-ceramic crowns with auxiliary retentive features on short clinical crowns. The findings demonstrate that CAD/CAM-fabricated crowns exhibit superior marginal and internal fit compared to conventionally fabricated crowns, with lithium disilicate crowns outperforming zirconia crowns in terms of adaptability. Additionally, crowns without grooves achieved the best marginal fit, highlighting the challenges associated with milling complex geometries. The volume of total cement space showed no significant variation among the groups, indicating that the cementation process was consistent across different crown types and designs. These results reinforce the potential of CAD/CAM technology as a reliable method for producing restorations with improved clinical performance in cases of short clinical crowns. CAD/CAM crowns with auxiliary grooves are recommended for clinical applications where retention is critical, provided the limitations of the technique are acknowledged and addressed.

**List of abbreviations**

| | |
|---|---|
| **3-D** | 3 Dimensional |
| **SCC** | Short clinical crowns/abutment |
| **CAD-CAM** | Computer-Aided Design and Computer-Aided Manufacturing |
| **PC** | Permanent crown |

| STL | Standard tessellation language |
| DICOM | Digital Imaging and Communications in Medicine |
| FDM | Fused deposition modelling |
| SLA | Stereolithography |
| SLS | Selective laser sintering |
| SLM | Selective laser melting |
| PBP | Powder binder printers |
| DLP | Digital light processing |
| CFL | Chamfer Finish line |
| RSFL | Rounded shoulder Finish line |
| RSBFL | Rounded shoulder with bevel Finish line |
| Z | ZONE |

### Funding
The authors received funding from the Deanship of Research and Graduate Studies at King Khalid University through Small Research Project under grant number RGP1/96/46. The funders had no role in study design, data collection and analysis, decision to publish, or preparation of the manuscript.

### Grant Disclosures
The following grant information was disclosed by the authors:
Deanship of Research and Graduate Studies at King Khalid University through Small Research Project: RGP1/96/46.

### Competing Interests
The authors declare there are no competing interests.

### Author Contributions
- Saeed M. Alqahtani conceived and designed the experiments, performed the experiments, analyzed the data, prepared figures and/or tables, authored or reviewed drafts of the article, and approved the final draft.
- Saurabh Chaturvedi conceived and designed the experiments, performed the experiments, analyzed the data, prepared figures and/or tables, authored or reviewed drafts of the article, and approved the final draft.
- Mohamed Khaled Addas conceived and designed the experiments, analyzed the data, prepared figures and/or tables, authored or reviewed drafts of the article, and approved the final draft.
- Nasser M. Alqahtani conceived and designed the experiments, analyzed the data, prepared figures and/or tables, authored or reviewed drafts of the article, and approved the final draft.
- Mohammad A. Zarbah conceived and designed the experiments, analyzed the data, prepared figures and/or tables, authored or reviewed drafts of the article, and approved the final draft.

- Mohammed Hussain Dafer Al Wadei conceived and designed the experiments, analyzed the data, prepared figures and/or tables, authored or reviewed drafts of the article, and approved the final draft.
- Feras Ali Alsaeed conceived and designed the experiments, performed the experiments, analyzed the data, prepared figures and/or tables, authored or reviewed drafts of the article, and approved the final draft.
- Yasir Saad AlJaadan conceived and designed the experiments, performed the experiments, analyzed the data, prepared figures and/or tables, authored or reviewed drafts of the article, and approved the final draft.
- Ali Abdullah Ali Alqahtani conceived and designed the experiments, performed the experiments, analyzed the data, prepared figures and/or tables, authored or reviewed drafts of the article, and approved the final draft.
- Mohammed Abdullah Al Mansooi conceived and designed the experiments, performed the experiments, analyzed the data, prepared figures and/or tables, authored or reviewed drafts of the article, and approved the final draft.
- Mudita Chaturvedi conceived and designed the experiments, analyzed the data, prepared figures and/or tables, authored or reviewed drafts of the article, and approved the final draft.

## Ethics

The following information was supplied relating to ethical approvals (*i.e.*, approving body and any reference numbers):

Institutional Review Board, College of Dentistry, King Khalid University.

## Data Availability

The raw data is available in the Supplemental File.

## Supplemental Information

Supplemental information for this article can be found online at http://dx.doi.org/10.7717/peerj.19813#supplemental-information.

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
