# Peer review of "Internal and marginal fit of digitally fabricated all-ceramic crowns with auxiliary retentive features on short clinical abutments: a micro-CT study"

_PeerJ, doi:10.7717/peerj.19813_

## Round 0.1 · original submission · Major Revisions

**Language Note:** The review process has identified that the English language must be improved. PeerJ can provide language editing services - please contact us at [email protected] for pricing (be sure to provide your manuscript number and title). Alternatively, you should make your own arrangements to improve the language quality and provide details in your response letter. – PeerJ Staff

·

Basic reporting

The manuscript is clearly written with unambiguous English, making the study easily understandable. The introduction and background sections are sufficiently developed, presenting a clear rationale for the research. However, there are areas where the manuscript could be improved:
Although current references adequately support your manuscript, incorporating recent literature (particularly studies published in 2023-2024) could significantly enhance the context and relevance. Updating references will better situate your work within the latest advancements in the field.
The manuscript generally conforms to PeerJ standards. However, clarity can be improved, especially in the methodology section. It is recommended to provide more detailed justifications regarding the specific CAD/CAM techniques selected, along with clearer explanations for choosing zirconia and lithium disilicate materials.
Figures and tables included are relevant and of high quality. However, it is recommended to provide more detailed legends explaining each measurement zone, explicitly linking them to corresponding text descriptions. This will enhance clarity and facilitate readers' understanding of the results.
The study represents a coherent body of work with a clear hypothesis. The manuscript is well-contained and does not appear to be artificially subdivided.
Addressing these points will significantly improve the manuscript, enhancing its clarity, scientific rigor, and overall suitability for publication.

Experimental design

The research presented is original, within the scope of PeerJ and addresses an important clinical question regarding dental restorations for short clinical crowns. The research question is well defined.
The methodological approach is reproducible and appropriate. However, there are areas for improvement:
It would be useful to clarify why specific CAD/CAM systems and materials (zirconia and lithium disilicate) were chosen. Further elaboration of the rationale behind these choices would strengthen methodological clarity.
Ethical approval, demonstrating compliance with research standards, has been sufficiently mentioned. However, further elaboration of the standardization procedures used during cementation, in particular the consistency of the applied finger pressure (why was it performed by more than one person and not appropriate inter-observer analyses?) would enhance methodological rigor.
While micro-CT scanning procedures are technically sound, detailing the criteria used to select measurement points and standardization methods during analysis would further strengthen reproducibility and technical precision.
Addressing these points would significantly increase the methodological transparency and overall scientific rigor of the paper.

Translated with DeepL.com (free version)

Validity of the findings

The findings presented in the paper are robust, supported by appropriate statistical analyses and clearly address the initial research question. The statistical analyses conducted (one-way ANOVA and post hoc Tukey tests) are appropriate and clearly demonstrate significant differences between the study groups.
The conclusions are well articulated, clearly related to the original research question and adequately limited to the results obtained. The article maintains scientific objectivity, avoiding exaggerated claims beyond what the data support.
Although the article presents findings that contribute meaningfully to the existing literature, it does not explicitly discuss how they add value beyond existing studies in similar contexts. Clearly articulating the rationale for performing this specific replication or variation and highlighting the specific contributions of this study to existing knowledge would strengthen the validity section.
Overall, the validity and conclusions of the paper meet high scientific standards and only minor improvements are suggested to clarify the unique contributions of the study.

Translated with DeepL.com (free version)

Additional comments

To enhance clinical applicability, consider briefly discussing potential implications of your findings for clinical practice.
Limitations of the study, particularly regarding in vitro conditions versus actual clinical scenarios, should be explicitly acknowledged to ensure readers understand the practical limitations clearly.
Future research recommendations based on your findings could provide valuable guidance for subsequent studies and help establish the next steps in this research area.

Reviewer 2 ·

Basic reporting

The manuscript is generally understandable and technically accurate in content. There are numerous grammar, syntax, and punctuation issues (e.g., inconsistent use of articles, awkward phrasing, typographical errors like “9” instead of apostrophes). Therefore, a thorough language edit is needed for fluency, professionalism, and readability.
The authors should consider tightening the introduction slightly to avoid redundancy and ensure consistent citation formatting. The authors should also reformat all citations, check for accuracy, and ensure each citation is correctly linked to a source in the references section.
In the methods, the blinding of operators is mentioned briefly but could be more clearly explained. Also, in the Methods section, the exact number of dies per group and how duplicates were assigned isn’t very clear. For the statistical analysis the authors should clarify whether normality and homogeneity of variance assumptions were checked before using parametric tests?

Experimental design

The study's aims align with prosthodontics, dental materials, and digital dentistry, all of which fall under most dental journals’ aims and scope. addresses a specific clinical challenge (short clinical abutments) and contributes novel insight by combining retentive groove geometry with CAD/CAM crowns—an area with limited prior investigation. The problem is linked to crown retention failure, a major clinical complication, especially in posterior restorations.
The authors should clarify how blinding, randomization, and cementation force were controlled; consider adding mention of inter-operator reliability in micro-CT readings, if available.

Validity of the findings

Although some elements like marginal gap analysis have been studied before, the combination with auxiliary features in SCC, CAD/CAM fabrication, and comparative material assessment (LiDS vs Zr) offers a valuable contribution. Key conclusions (e.g., differences in MG and IF) are backed by statistically significant findings (P < 0.001), while non-significant results (like cement volume) are appropriately discussed.

Additional comments

This manuscript presents original research that contributes meaningfully to the literature on digital prosthodontics and fixed restorations. The scientific question is clearly defined, the methodology is robust, and the analysis is appropriate.

In light of these observations, the following recommendations are proposed:
major revision – The manuscript is suitable for publication upon addressing the comments related to language polishing, methods clarification, and data presentation.

---

## Round 0.2 · accepted · Accept

Dear authors, we are pleased to verify that you meet the reviewer's valuable feedback to improve your research.
Thank you for considering PeerJ and submitting your work.

Kind regards
PCoelho

Reviewer 2 ·

Basic reporting

-

Experimental design

-

Validity of the findings

-